# Mediating Effect of Self-Control on the Relationship between Obesity-Related Stress and Weight Control Behavior among Female College Students with Overweight and Obesity

**DOI:** 10.3390/healthcare12050522

**Published:** 2024-02-22

**Authors:** Young-Hee Park, Yeo-Won Jeong, Hyun-Kyeong Park, Seul-Gi Park, Ha-Yeon Kim

**Affiliations:** 1Department of Nursing, College of Nursing, Dongguk University, Gyeongju 38066, Republic of Korea; gml34@dongguk.ac.kr (Y.-H.P.); ywjeong@dongguk.ac.kr (Y.-W.J.); 2Department of Nursing, Graduate School, Dongguk University, Gyeongju 38066, Republic of Korea; inau1295@naver.com (S.-G.P.); 2230013@hyumc.com (H.-Y.K.)

**Keywords:** overweight, obesity, self-control, psychologic stress, behavior

## Abstract

Background: This study aimed to determine whether self-regulation acts as a mediating factor in the relationship between obesity-related stress and weight control behavior, targeting Korean female college students. Obesity-related stress and weight control behaviors are important factors affecting an individual’s health and well-being, and self-control, a psychological trait, can play a pivotal role in regulating how individuals respond to stress and engage in weight management behaviors. Methods: This study employed a descriptive correlational design. A total of 109 female college students with overweight and obesity participated in the study. We used a self-report questionnaire to measure obesity-related stress, self-control, and degree of weight control behavior. Data were analyzed using descriptive statistics, Pearson’s correlation coefficient analysis, and bootstrapping using Hayes’ PROCESS macro for mediation. Results: Significant relationships between obesity-related stress and weight control behavior (r = 0.25, *p* < 0.001), obesity-related stress and self-control (r = −0.36, *p* < 0.001), and weight control behavior and self-control (r = 0.26, *p* < 0.001) were observed. Self-control showed an indirect mediating effect on the relationship between obesity-related stress and weight control behavior (Β = 0.37, *p* = 0.001). Conclusions: Through the research results, it was confirmed that self-control is a factor that has a mediating effect in the relationship between obesity-related stress and weight control behavior among female college students with overweight and obesity. These results underscore the importance of addressing self-control strategies in interventions aimed at promoting healthy weight management among female college students with overweight or obesity.

## 1. Introduction

Obesity is a known risk factor for various physical as well as psychological disorders [1], and the incidence of health problems begins to increase at a body mass index (BMI) of 23–27 kg/m^2^ [2], highlighting the need for weight control at an overweight stage. In 2022, the adult obesity rate in Korea was relatively low, at 46.1% for men and 29.6% for women, compared with the average rates (male: 62.9%; female: 54.5%) in 35 Organization for Economic Co-operation and Development (OECD) countries [3]. However, the rate of increase in cases of obesity in Korea is very steep compared to Western countries; moreover, the OECD projects the rate of severe obesity rate in Korea to reach 9.0% in 2030, approximately two-fold higher than that in 2015 [4]. 

An analysis of adults with obesity rates from 2008 to 2021 found that obesity increased in all age groups for men, while the increase was more pronounced for women in their 20s and 30s. [5]. The background of this phenomenon is thought to be related to research results that reported that increased sedentary behavior among college students due to study and job preparation leads to non-physical activity and is prone to overweight and obesity due to an imbalance between energy intake and consumption [6]. In fact, the prevalence of Korean female college students with obesity is rapidly increasing, and this is a major health problem [7]. Apart from its physical ramifications, obesity often precipitates psychological distress, including stress related to body image and weight management. This obesity-related stress can influence individuals’ weight control behavior, impacting their efforts to maintain a healthy lifestyle [8].

Individuals with overweight or obesity have been reported as a risk factor for unhealthy weight control behaviors (UWCB) [9]. Female students, in particular, often experience dissatisfaction with their bodies and express concerns, which negatively impact their health [10]. UWCB refers to certain behaviors that are generally not recommended for weight management, such as skipping meals, eating little, fast, taking diet pills, vomiting, using laxatives, using diuretics, and smoking [11]. This phenomenon is attributed to a societal emphasis on external appearance, especially as students prepare to graduate and enter the workforce [12]. For women, there has been a significant increase in media pressure to achieve thin/low-fat ideals, resulting in the majority of students perceiving themselves as obese and experiencing significant obesity-related stress [13,14]. It is reported to be higher in women compared to men, influenced by excessive focus on obesity and past experiences with weight control, and prevalent among individuals with obesity and a high BMI [15,16]. Foss and Dyrstad [17] reported that obesity and stress are mutually influential, and research focused on obesity-induced stress is warranted for appropriate weight loss and enhancement of the quality of life [18]; While numerous studies have explored the relationship between obesity-related stress and weight control behavior, the role of self-control as a mediator remains underexplored, particularly among the college student population.

Self-control is the determination and ability to control one’s inner impulses and behaviors to attain one’s own standards or goals [18,19]. It is associated with behaviors in various aspects, such as diet, interpersonal relationships, planning, and decision-making [20], and is reported as an important factor in facilitating desirable health behaviors [21]. In terms of obesity, self-control contributes to controlling BMI [22] and weight control behaviors [23]. While individuals with good self-control are able to steadily and appropriately manage their weight by ameliorating their lifestyle, those with poor self-control exhibit impaired impulse control and engage in undesirable eating behaviors [23]. Self-control is also linked to psychological adaptation [19]. Excessive stress can impair self-control [24], and it plays an important role in suppressing undesirable impulses and maintaining positive behaviors through regulating negative mood or emotions [18]. Based on these findings, self-control is anticipated to have a mediating effect on the relationship between obesity-related stress and weight control behaviors in overweight or obese female college students with a high BMI; however, no previous study has investigated the effects and relationship among the three variables. 

Through this study, it will be possible to confirm the presence or absence of a mediating effect of self-control in the relationship between obesity-related stress and weight control behavior experienced by overweight and obese female college students. The results of this study will be an initial study confirming the role of self-control in inducing healthy and sustainable weight control behaviors among young people with overweight and obesity. Additionally, if the mediating effect of self-control is confirmed, it is expected to be important evidence that self-control should be considered to induce healthy weight control behavior in individuals with overweight and obesity in the future. Thus, this study aimed to investigate the relationship among obesity-related stress, weight control behavior, and self-control in female college students with overweight or obesity.

## 2. Materials and Methods

### 2.1. Design of the Study

This study was a descriptive cross-sectional survey performed using a self-reported questionnaire to investigate the degree of obesity-related stress, weight control behavior, and self-control of overweight and obese female college students and confirm the mediating effect of self-control in the relationship between obesity-related stress and weight control behavior. 

A primary objective of this study was to examine the mediating effect of self-control between obesity-related stress and weight control behavior in female college students with overweight or obesity. Additionally, the authors were interested in understanding the effects of obesity-related stress and self-control on weight control behaviors in female college students with overweight or obesity and the associated outcomes.

### 2.2. Participants in the Study

The participants of this study were surveyed through convenience sampling of female students attending universities located in two metropolitan cities (Seoul and Ulsan) and two provinces (Gyeongbuk and Chungnam). This study included female students older than 18 years with a BMI of 23 kg/m^2^ or higher who were currently in university, understood the purpose of this study, and agreed to participate in it. A G*power of 3.1.9.2 [25] was used for sampling; the minimum number of participants was 109 when the significance level was set at 0.05, while the size of the effect was set at 0.15, and the statistical power of the test was set at 0.80 when the impact factors were set at 8. Thus, a total of 120 participants were sampled, considering a dropout rate of 10%.

### 2.3. Study Tools

A structured questionnaire was used, which consisted of 46 items in total: 5 items about general characteristics, 15 about obesity-related stress, 11 about self-control, and 15 about weight control behavior.

#### 2.3.1. Obesity-Related Stress

To measure obesity-related stress, the scale developed by Jang Je-hyun and Shin Gyu-ok [26] for their research on obesity-related stress in female university students was used. The tool has been validated and consists of 3 areas, each with a total of 15 questions: 5 questions on effort stress (“trying to eat less”, “currently trying to lose weight by dieting”, etc.), 5 questions on psychological stress (“I’m afraid of gaining weight”, “I don’t feel confident when meeting other people”, etc.), and 5 questions on physical stress (“I feel fat when I take pictures”, “I often step on the scale to check my weight”, etc.), with a minimum score of 15 and a maximum score of 75. Each item was scored on a 5-point Likert scale (1 = not at all; 5 = a lot), which indicates that the higher the total score, the higher the obesity-related stress. In Jang Je-hyun and Shin Gyu-ok’s research [26], Cronbach’s α was 0.91, while in this study, it was 0.81.

#### 2.3.2. Self-Control

In order to measure the participants’ self-control, this study used a Brief Self-Control Scale (BSCS) of 13 questions developed by Tangney et al. [18] and translated and adapted by Hong Hyun Gi et al. [27]. In the process of validating this tool, two of the existing 13 questions were deleted by the results of the internal consistency and exploratory factor analysis between the questions, and the tool consisted of the final 2 areas and 11 questions. The first factor is self-control, for example, “I don’t give in to temptation easily, or I find it hard to break bad habits”. The second factor is focus, which includes statements such as “I am lazy and I can work efficiently towards long-term goals”, all measured on a 5-point Likert scale (1 = not at all; 5 = a lot). 

Nine out of 11 questions were calculated as reverse questions, and the total score was distributed between a minimum of 11 points and a maximum of 55 points, which indicates that the higher the total score, the higher the self-control. In Hong Hyun-Gi’s research [27], Cronbach’s α was 0.78, while in this study, it was 0.82.

#### 2.3.3. Weight Control Behavior

Weight control behavior was assessed using the scale developed by Jeong Hui-seop [28] and modified and balanced by Jeong Yun-kyung and Tae Young-sook [29]. The scale has been validated and consists of 4 areas with 15 items, with measurement on a 5-point Likert scale (1 = not at all; 5 = a lot): 3 about exercise therapy (“exercise, such as jogging, gymnastics, or swimming”, etc.), 8 about dietary therapy (“eat a balanced diet”, etc.), 1 about drug therapy (“take medications to lose weight, such as diuretics, laxatives, and herbal medicines”), and 3 about behavioral therapy (“weigh yourself regularly and watch for changes”, etc.). The total score was between 15 and 75, which indicates that the higher the scores, the more active the participants’ weight control behavior. The Cronbach’s α was 0.77 in Jeong Yun-kyung and Tae Young-sook’s research [29] and 0.79 in this study.

### 2.4. Ethical Consideration

This study was examined and approved by the Institutional Review Board of the researcher’s institution, Dongguk University (DGU IRB 20200032). On the first screen of the online survey, the contents, including anonymity, voluntary participation, and rejection, when participating in the study with ethical considerations, withdrawal during the study, and possible benefits and disadvantages, were described. In addition, it was announced that their mobile phone numbers would only be used as a reward for responses and would be immediately disposed of after this study. The participants were free to decide regarding their participation in this study when they individually reviewed the first screen, and the collected data were immediately coded by the provision of an ID to conceal the identity of the participants to prevent anonymity. To take part in the study, participants were given a $5 coffee voucher by 50 people in a lottery.

### 2.5. Data Collection

The data were collected using a structured self-reporting questionnaire for the period from 18 November 2020 to 30 November 2020 online from female university students in Seoul, Ulsan, Gyeongbuk, and Chungnam. To collect data, three researchers posted the addresses of the online survey at their school and nearby schools. The first screen of the survey described the background, purpose, methods, ethical considerations, and standards for the selection of the participants. If the participant consented to participating in the study, she was asked to fill out “Yes” or “Agreed” at the bottom. The upper part of the second screen contained the definition and calculation method of BMI (weight [kg] ÷ height [m^2^]). The participants could calculate their BMI indices personally and input them. If the index was 23 kg/m^2^ or higher, they would be guided to press the ‘Next’ button to participate in the survey.

### 2.6. Data Analysis

Data were analyzed using the SPSS/WIN 25.0 program (IBM Corp., Armonk, NY, USA) and SPSS/WIN PROCESS macro v3.4. First, general characteristics and main variables were processed for frequency, percentage, means, standard deviations, and other descriptive statistics. In addition, all main variables satisfied the assumption of normality (skewness: −0.178 to 0.123, kurtosis: −0.139 to 0.232). Second, the correlations among the participants’ obesity-related stress, self-control, and weight control behavior were analyzed using Pearson‘s correlation coefficients. Third, the SPSS PROCESS macrol (model 4) was used to analyze the mediating effects of self-control on the relationship between the participants’ obesity-related stress and their weight control behavior. In order to confirm the statistical significance of the mediating effects, Hayes’ PROCESS macro was used for boot strapping and reported the adjusted OR. Also, this study declared a significance level of *p* < 0.05 and a 95% CI.

## 3. Results

### 3.1. Simple Characteristics

A total of 120 questionnaires were collected; 110 questionnaires were used for the final analysis, whereas 10 copies that could not be used for analysis owing to insufficient data (partial or insincere responses) were excluded. Participants’ average age was 23.21 years, while their average BMI was 25.80 kg/m^2^. Specifically, 58.2% of them were in their 4th year of university, and 90% of them had tried to control their weights within the last year. A total of 74.5% (n = 82) of them consumed alcohol less than twice per week, while 90% of them did not smoke (Table 1).

### 3.2. Research Hypothesis Validation

#### 3.2.1. Correlations among the Participants’ Obesity-Related Stress, Self-Control, and Weight Control Behavior

The results of research questions 1 and 2 are shown in Table 2. The participants’ average obesity-related stress score was 45.05 ± 9.11, while their average self-control and their weight control behavior scores were 32.82 ± 7.13 and 42.20 ± 8.27, respectively (Table 2). The participants’ obesity-related stress positively correlated with their weight control behavior (r = 0.25, *p* < 0.001), while negatively correlated with their self-control (r = −0.36, *p* < 0.001). Their self-control positively correlated with their weight control behavior (r = 0.26, *p* < 0.001) (Table 2).

#### 3.2.2. Mediating Effects of Self-Control in the Relationship between Obesity-Related Stress and Weight Control Behavior

To examine the mediating effect of self-control on weight control behavior, the following values were controlled for covariates (age, grade, experience of weight control, alcohol, and smoking). Considering the relationship between obesity-related stress and weight control behavior, as in research question number 3, self-control had a partial mediating effect (Figure 1). 

Obesity-related stress had negative effects on self-control (Β = −0.26, *p* < 0.001), while weight control behavior had positive effects (Β = 0.36, *p* < 0.001). In addition, self-control had some positive effects on weight control behavior (Β = 0.37, *p* = 0.001) (Table 3). Considering the relationship between obesity-related stress and weight control behavior, the indirect effects of self-control were significant. (Β = −0.09, bootstrap 95% confidence interval: −0.188, −0.02).

## 4. Discussion

In this study, the obesity-related stress score was higher than the score among female college students reported by Kang and Kim [30] using the same instrument. The difference in the obesity-related stress scores seems to be attributable to the fact that we included female college students with a BMI of 23.0 kg/m^2^ or higher, while Kang and Kim [30] did not use a BMI criterion for their participants. Moreover, female students with overweight or obesity and a BMI of 23.0 kg/m^2^ or higher are likely to experience higher levels of obesity-related stress, supporting Jeong’s [16] findings that those with higher obesity may be more dissatisfied with their body shape and have more obesity-related stress.

There was a significant positive correlation between obesity-related stress and weight control behavior. This is similar to the report by Yang and Byeon [31] that the perceived overweight and normal weight groups more frequently engage in weight control behaviors than the perceived underweight group. Further, this is consistent with previous findings showing a significant association between weight control behaviors and obesity-related stress in female college students [15,32]. Thus, obesity-related stress may serve as a motivator for weight control behaviors as individuals become aware that they are obese on their own (internal stimulation) or through people around them or mass media (external stimulation) [15]. However, some studies also report that weight control behaviors induced by obesity-related stress are often undesirable weight loss attempts, where individuals try to lose weight in a short period of time regardless of attempts to promote health, which eventually leads to failed attempts [30,33]. Therefore, studies should examine the predictors of weight control behaviors other than obesity-related stress, and replication studies as well as studies analyzing the associations among and the effects of various positive variables and weight control behaviors are required. 

Throughout this research, self-control showed a significant partial mediating effect in the relationship between obesity-related stress and weight control behavior. The association between self-control and objective measures of adiposity has rarely been investigated in young people [34]. In this regard, a study by Boat et al. [34] reported that self-control may be an attractive target for future interventions to reduce obesity because it is related to healthy behaviors and characteristics in adolescents. This is a result in a similar context to the results of this study. Additionally, previous studies have shown that multidisciplinary obesity treatment (MOT) already includes general self-control training for childhood obesity [35]. And self-control training for weight loss has been reported as a promising pathway to improve the long-term outcomes of multidisciplinary obesity treatment [36]. Therefore, the results of this study can be presented as evidence that self-control can be a psychological antecedent or personal factor that can help develop more effective preventive strategies to lead to healthy weight control behaviors.

Considering the mediating effects, our results showed that self-control influences weight control behaviors in female college students, suggesting that improving self-control to ensure that individuals do not choose unhealthy weight control methods is essential in addition to an appropriate level of obesity-related stress in order to motivate female college students with overweight and obesity to engage in positive weight control behaviors. This is supported by the study on academic stressors and cyberloafing in college students by Zhou et al. [37], where the association between the two factors was significant in students with poor self-control, but students with high self-control had a low risk of cyberloafing regardless of the academic stressors. Thus, it is necessary to explore other potential mediators, such as internalized stigma and negative self-perceptions [38], that influence weight control behavior in order to understand the effects of obesity-related stress on weight control behavior.

## 5. Strengths and Limitations of the Study

This study has several limitations. Although we recruited female college students from different regions through an online survey, the criterion of a BMI of 23.0 kg/m^2^ or higher meant that we only recruited a minimal sample. Further, there could be errors with the BMI measurements since they were measured individually using their weight and height by themselves. Thus, the findings of this study should be generalized with caution, and future studies should ensure accuracy in the selection of female college students with overweight and obesity by employing objective measurements of BMI. In addition, since this study was conducted only on female college students with overweight and obesity, further studies are recommended to determine whether there are differences in obesity-related stress and variables affecting weight control behavior according to gender.

Despite the limitations of the study, this study confirmed the importance of self-control in female college students with overweight and obesity as the first paper to verify the mediating effect of self-control between obesity-related stress and weight control behavior. In addition, the results of this study highlight the need for schools and relevant organizations in the community to provide education and counseling for female college students with overweight and obesity to enhance self-control such that they can choose positive weight control behavior, and these education and counseling programs should include contents pertaining to sustained appropriate diet, exercise therapy, and stress-coping skills for the development of a desirable body image and better health promotion. These results would bolster the understanding of the psychosocial factors and weight control behavior of female adolescents and female college students in school health facilities and by clinical nurses, thereby improving their support for students’ weight control efforts.

## 6. Conclusions

This study aimed to investigate the mediating effect of self-control on the relationship between obesity-related stress and weight control behavior in female college students, and the results confirmed that obesity-related stress was significantly correlated with weight control behavior and self-control, while weight control behaviors were significantly correlated with self-control. Self-control partially mediated the relationship between obesity-related stress and weight control behavior in female college students. Thus, self-control resulted in the evolution of obesity-related stress into weight control behaviors. Hence, programs and strategies are needed to enhance self-control in order to reduce the prevalence of overweight and obesity among female college students. Moreover, an appropriate level of stress from obesity helps facilitate weight control behaviors; thus, programs to manage obesity-related stress should be developed and implemented to prevent the progress of such stress to aggressive weight loss attempts in a short period of time.

## Figures and Tables

**Figure 1 healthcare-12-00522-f001:**
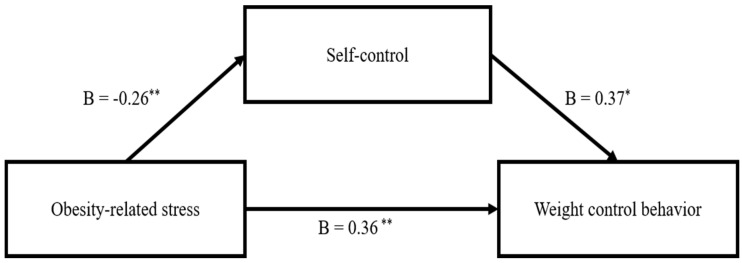
Mediating effect of self-control on the relationship between obesity-related stress and weight control behavior (* *p* < 0.01, ** *p* < 0.001).

**Table 1 healthcare-12-00522-t001:** General characteristics (n = 110).

Variables	Mean ± SD	Range	N (%)
Age (years)	23.21 ± 1.95	20–33	
Body mass index	25.80 ± 2.92	23–36	
Grade			
Freshman			7 (6.4)
Sophomore			11 (10.0)
Junior			28 (25.5)
Senior			64 (58.2)
Experience an attempt to control weight during the last year			
Yes			99 (90.0)
No			11 (10.0)

SD, standard deviation.

**Table 2 healthcare-12-00522-t002:** Correlations among the main variables (n = 110).

	Mean ± SD	Range	1	2	3
1. Obesity-related stress	45.05 ± 9.11	21–66	1		
2. Self-control	32.82 ± 7.13	13–52	−0.361 **	1	
3. Weight control behavior	42.20 ± 8.27	21–64	0.254 **	0.263 **	1

SD, standard deviation; ** *p* < 0.001.

**Table 3 healthcare-12-00522-t003:** Mediation testing results (n = 110).

Outcome		Β	SE	*p*	LLCI	ULCI
	Constant	58.57	9.99	<0.001	38.74	78.40
Self-control	Obesity-related stress	−0.26	0.07	<0.001	−0.41	−0.11
		R^2^ = 0.22, F = 4.19, *p* < 0.001				
	Constant	27.91	12.84	0.03	2.43	53.40
Weight control behavior	Obesity-related stress	0.36	0.08	<0.001	0.18	0.53
	Self-control	0.37	0.11	0.001	0.15	0.59
		R^2^ = 0.29, F = 5.25, *p* < 0.001				

LLCI, lower limit confidence interval; ULCI, upper limit confidence interval.

## Data Availability

The data presented in this study are available on request from the corresponding author. The data are not publicly available to protect the privacy of human subjects.

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
