# Peer review of "Mediating Effect of Self-Control on the Relationship between Obesity-Related Stress and Weight Control Behavior among Female College Students with Overweight and Obesity"

_healthcare, 2024, doi:10.3390/healthcare12050522_

Round 1

Reviewer 1 Report

Comments and Suggestions for Authors

Interesting study that investigates relationships between obesity and internal health and weight behaviors.  The study would be improved with significant revisions: 

A few notes:  Obesity is a complex chronic disease that includes the individual, environmental, genetic, metabolic and psychosocial factors.  Additionally, person-centered language is recommended when writing and discussion obesity.  For example:  " female college students with overweight and obesity"  vs.  "overweight and obese female college students".  The authors also need to state the complexity of obesity as a medical condition and disease, and should cite more recent peer review articles related to obesity- in general, and the multiple factors involved.  

It is unclear what are the objectives of this study?  Are the authors trying to investigate individual and psychological/psychosocial aspects of obesity and how they are correlated with one another?  

Specific recommendations:  

1.  reword non-person centered statements throughout the paper:  For example in the Abstract :  "This study aimed to examine the obesity stress, weight control behavior, and 13 self-control in overweight and obese female university students"- should be restated: "....in female university students with overweight and obesity".   This phrasing should be used throughout the paper

2.  Introduction:  line 34 "morbid obesity" is an outdated term - and should be "severe obesity" 

3. Review of Literature"  

-lines 40-41"In general, women have a higher percent body fat percentage than men and thus are  more likely to progress towards obesity" This statement needs to be justified with recent peer review articles.  

-lines 42-43 recommend rewording  to ""may attempt to lose weight in a way that impairs health"

-lines 44-48 "Further, a majority of college students perceive themselves as obese and suffer from severe obesity  stress [8]. Obesity-related stress refers to perceived stress due to obesity [9], which is reportedly influenced by excessively focus on obesity and past weight control experiences  and is considered higher among women than men and is highly prevalent among obese individuals with a high BMI [10, 11].** More than 1 citation is needed to justify the first sentence.  How is 'majority' defined?  Obesity-related stress refers to perceived stress due to obesity By this definition- the authors are saying that 'the majority' of female university students have obesity?  Please revise- this is a run-on sentence and confusing to the reader:  Obesity-related stress refers to perceived stress due to obesity [9], which is reportedly influenced by excessively focus on obesity and past weight control experiences  and is considered higher among women than men and is highly prevalent among obese individuals with a high BMI [10, 11].

-lines 59-61 "While individuals with good self-control are able to steadily and appropriately manage their weight by ameliorating their lifestyle, those with poor self-control exhibit impaired impulse control and engage in undesirable eating behaviors that eventually result in psychosocial health problems" **Recent and more peer review articles are needed to support this statement. Currently- this reads as opinion.   In the obesity field (prevention and treatment) , obesity is recognized as a disease- that is complex- with various factors:  environmental, genetic, social, psychosocial, and individual.  While self-control may be an aspect of obesity, this currently reads as if self-control is the only factor.

4. Methods:  currently the only validated study tool/questionnaire used was for assessing self- control.  What about the other 2 surveys? (for obesity stress and for weight control behavior?  It would be helpful for example surveys to be included.  Please justify the use of each questionnaire.  

4. Results:

-Table 1 needs to be revised.  "General" in the title of table is misspelled; what does "grade" mean?  Please explain " Experience of attempted to control weight during last..."  is this a timeframe?  If so- this should be represented more clearly.  why is alcohol included in this section?  This seems out of place. 

5.  Discussion: 

lines 247-251:  Self-control is the ability to stop undesirable behavioral inclinations, such as impulses, and refrain from relevant behaviors and a psychological variable that can reduce the impact of stress [13, 31]. Hence, many studies have demonstrated that self-control  predicts better adjustment and better performance [15, 32]. Therefore, individuals with  high self-control have a sense of duty and are disciplined and goal-oriented [33].  These statements do not seem relevant to obesity- and it seems the authors are attempting to make a correlation between self- control and obesity.  Please rework this, and discussion "self- control" within the context of current obesity literature.  

Comments on the Quality of English Language

Editing of paper is required. 

Author Response

Dear Reviewer:

I wish to submit an article for publication in Healthcare, titled “Mediating effect of self-control on the relationship between obesity-related stress and weight control behavior among female college students with overweight and obesity” The paper was coauthored by Young-Hee Park, Yeo-Won Jeong, Hyun-Kyeong Park, Seul-Gi Park and Ha-Yeon Kim.

The purpose of this study is to investigate Mediating effect of self-control on the relationship between obesity-related stress and weight control behavior among female college students with
overweight and obesity. The novelty of this study is its focus on exploring the effects of self-control in individuals with overweight and obesity, and specifically filling the gap in the literature examining the effects of self-control among weight control behavior in young people with overweight and obesity. Policy implications may include emphasizing that medical institutions and medical professionals must consider self-control as important in order to induce healthy
weight control among individuals with overweight and obesity, a public health issue that is increasing worldwide. We believe that our study makes a significant contribution to the literature because the positive findings can be incorporated into nursing practice and as part of school or workplace health programs.

This manuscript has not been published or presented elsewhere in part or in entirety and is not under consideration by another journal. All study participants provided informed consent, and the study design was approved by the appropriate ethics review board. We have read and understood your journal’s policies, and we believe that neither the manuscript nor the study violates any of these. There are no conflicts of interest to declare.

Thank you for your consideration. I look forward to hearing from you.

Sincerely,
Hyun Kyeong Park
Department of Nursing, College of Nursing
Dongguk University Gyeongju Campus
123, Dongdae-ro, Gyeongju-si
Gyeongsangbuk-do, Korea 38066
Tel. 82-10-4286-3678
Fax. 82-54-770-2616
E-mail: sg3679@dongguk.ac.k

Reviewer 2 Report

Comments and Suggestions for Authors

There are a number of issues and questions for the authors.

1) Line 13 and 15 - 'obesity stress' is misleading and undefined in the abstract. Therefore, please refer to as 'obesity-related stress.'

2) Please refrain from describing individuals as 'overweight and/or obese.' Standard practice is to avoid stigmatizing language and use People First Language (see official statements from Obesity Action Coalition and others) and refer to 'individuals with overweight' or 'individuals with obesity.'  There are 19 instances, each, of the terms overweight and obese in the manuscript that need to be changed. 

3) Line 28. Obesity is not defined as an accumulation of fat. Obesity is driven by inflammation and hormonal dysregulation (see statemtents by The Obesity Society and the American College of Endrocrinology, among others). Please restate this. 

4) Line 47. I think you mean "excessive focus."

5) Line 49. Obesity and stress are mutually influential. This is true, but incomplete. The stress is in turn both psychological and physiological (driven by cortisol, etc). The authors also fail to address the influence of stigma and both external and internal weight bias.  This literature and discussion needs to be included. 

6) Line 74. Is this questionnaire validated?

7) Line 78. These are not research questions. These are hypotheses. Please correct one way or another. 

8) Line 90. Reference for G*Power please.

9) Line 99. Obesity-related stress.

10) Line 109. Was it validated after translation and adoption?

11) Results. These results are interesting, but without accounting for stigma and bias effects, I fail to see how correlations add to our knowledge or understanding. 

12) Line 275-280. The authors state the results are not generalizable. Agreed. But then in these lines the authors call for school-wide and community-wide action to be taken based on these results. Whereas, this is not arguable, how do the authors account for the lack of generalizability leading to such robust recommendations? 

Comments on the Quality of English Language

Please see general comments to authors. Otherwise, acceptable. 

Author Response

(The authors gave the same response as above.)

Reviewer 3 Report

Comments and Suggestions for Authors

Abstract:

A brief opening sentence to provide insight into the rationale behind the project in the “background” sub-section is needed.

The scales used in the study need to be outlined in this section there to give context to the results being outlined.

Concluding remarks need to match the specific findings observed / reported in the study.

Introduction:

Section 1 and 2 can be combined into one.

Place L35-L38 after the semicolon in L29. Structurally this will flow a lot better.

Some insight as to and how why the obesity rates in Korea are higher than the west needs to be explained and explored (L33-34). This is rather important for the rationale development of the work given statements made in L51-53 (that studies in Korean women are lacking). What would measuring Korean individuals in particularly add? Do they have different obesity-related stressors for instance? Very vague arguments.

Line 43-44 – requires a reference and needs to be reworded for clarity (what is meant by this?).

More discussion and connection are required between obesity stress and self-control. Large literature omissions (for instance, there has little to appreciation of concepts like emotional eating and so on.  

In general, there has been very little outline of why this study is needed and what it will add. This section requires a full redraft. With clear insight to the rationale and context provided as to what it will add to understanding of the field. As weight control behaviours have been measured in the study, these have been largely ignored in this section. This should be addressed.  

Method:

3.2. Research Questions – this needs to be at the end of the introduction section. These also needed to be developed better within the introduction section itself.

It might be worth noting that being overweight (the BMI cut off) in Korea is different to the western measures of >25.

I couldn’t find the obesity stress scale (or it isn’t accessible) – reference 20. Is there any way to do this? As it is hard to understand the scale and questions being asked with out easy access. It isn’t a well-known / used scale. Perhaps provide an example of the question for each of the three suggested sections.

Are validity statistics available for these three scales?

Can the Cronbach alpha analysis for each scale be reported in the appendix / supplementary materials?

L137 - What is meant by data leakage?

What was the reward? Monetary reward? How much? Be specific.

What was the advertisement procedure of the survey?

Was there any data cleaning processes? Check for outliers etc.?

Results:

“Insufficient data” – does this mean incomplete data sets?

What does grade 1-4 indicate in Table 1?

Wording like “negative effects” is a miscommunication of the analysis that has been conducted. The correlations conducted are relationships not effects. These will need to be amended throughout.

Discussion:

Start this section with an outline of what the results were without numbers and statistics. Just a clear outline of what was found in words is needed.

L206 – where was the analysis of this in the results section? What is its significance to the 3 key research questions set out? Contextualize in light of these.

L252-255 – rather bold claim.

L265 – this criticism doesn’t make sense in the light of earlier comments of conducting power analysis. Basic errors.

It is important to remember that weight control behaviours (as a measure) are a proxy of a proxy outcome. It is unwise and rather bold to state that the results show self-control directly links to obesity. The connection between weight control behaviours and self-control is needed in the introduction to further explore here.

There is a vast literature connecting self-control and behaviours that lead to obesity. What has this study added that isn’t already known?

Comments on the Quality of English Language

Proofread the document thoroughly (a small number highlighted below – too many to list). There is a number of grammatical and poor writing errors throughout.

Opening L13 – “to examine the obesity stress” – doesn’t make sense.

Line 14 – “and identify the mediating effect” – and to identify.

Line 28 – incorrect use of semicolon.

L193 – “higher” self-control “scores” were positively correlated with…

L195 – relationship not relation.

Author Response

(The authors gave the same response as above.)

Round 2

Reviewer 1 Report

Comments and Suggestions for Authors

The paper is improved and thank you to the authors for reworking the writing to include more person centered language for obesity and overweight.  To even further improve the quality of the paper, please see the following recommendations below: 

Abstract.  Line 13: would reword the first sentence.  Are the authors trying to say that individuals with obesity perceive stress?  If so- this isn't a strong background statement  Recommend a congruent opening statement that supports the study's objectives/purpose. 

Introduction:  Lines 33-34.  Please reword the first sentence.  It sounds like the authors are trying to say that Obesity and Overweight, as a medical condition/disease is a growing public health challenge.  vs. the individuals themselves being a challenge

Line 44 "adults with" vs. "adult with". 

Line 54:  "the lower amount of physical activity, which can cause overweight and obesity"-  please rewrite this statement.  Unless the citation specifically notes causation (not likely), it will be most accurate to note a relationship.  A relationship/association is not causation. 

Line 58 Do the authors mean to say "negatively impact" in this statement?   " inappropriate weight-loss attempts that may impact their health [12]."

Materials/Methods: 

2.2 "Research hypothesis" reads too much like an informal section of a dissertation/thesis.  Please rewrite in complete sentences, and appropriate for a peer review journal article.   Example:  "A primary objective of this study was...  Additionally, the authors were interested in understanding outcomes related to..."    

For lines 106-111 please change this language to read " female college students with overweight or obesity". 

2.4.2-  lines 143-144.  Are these questions directly from the survey? If so- please put these questions in quotation marks".  Also- it is recommended that the actual survey questions be place in a Table. 

Figure 1:  please define "B"

Table 1: should read "Experience of attempt to control weight during the last 1 year"  (vs. 'attempted") 

Table 2 is confusing.  What do the "1, 2 "3 columns mean on the first row of the Table?  It looks like authors are trying to show correlations and based on scores from the surveys.  It would be more clear to the readers what the 'ranges' of scores mean?  This needs to be better explained in the methods, or by providing a Table with the survey questions.  

Table 3:  what does 'constant' refer to? would better define this in your methods, and in the footnote of your table.  Is this the average score for each outcome?  and based on the survey?  

Discussion: Line 244:  please change to "female students with overweight or obesity and with a BMI..." Would also reword "..findings that the more obese the person"-  with something like " findings that those with higher obesity may have more dissatisfaction with body shape and with more obesity related stress "..

Comments on the Quality of English Language

Please review the paper to improve the quality of English Language. 

Author Response

[18 Feburary 2024]

Dear Reviewer

I wish to submit an article for publication in Healthcare, titled “Mediating effect of self-control on the relationship between obesity-related stress and weight control behavior among female college students with overweight and obesity” The paper was coauthored by Young-Hee Park, Yeo-Won Jeong, Hyun-Kyeong Park, Seul-Gi Park and Ha-Yeon Kim.

Through this second review, we have made our best efforts to faithfully carry out the reviewer's comments. We sincerely appreciate you for your thoughtful and warm review, which helped us improve our study. The content modified in the second review is marked in blue color to make it easier for you to check.

The purpose of this study is to investigate Mediating effect of self-control on the relationship between obesity-related stress and weight control behavior among female college students with overweight and obesity. The novelty of this study is its focus on exploring the effects of self-control in individuals with overweight and obesity, and specifically filling the gap in the literature examining the effects of self-control among weight control behavior in young people with overweight and obesity. Policy implications may include emphasizing that medical institutions and medical professionals must consider self-control as important in order to induce healthy weight control among individuals with overweight and obesity, a public health issue that is increasing worldwide. We believe that our study makes a significant contribution to the literature because the positive findings can be incorporated into nursing practice and as part of school or workplace health programs.

This manuscript has not been published or presented elsewhere in part or in entirety and is not under consideration by another journal. All study participants provided informed consent, and the study design was approved by the appropriate ethics review board. We have read and understood your journal’s policies, and we believe that neither the manuscript nor the study violates any of these. There are no conflicts of interest to declare.

Thank you for your consideration. I look forward to hearing from you.

The purpose of this study is to investigate Mediating effect of self-control on the relationship between obesity-related stress and weight control behavior among female college students with overweight and obesity. The novelty of this study is its focus on exploring the effects of self-control in individuals with overweight and obesity, and specifically filling the gap in the literature examining the effects of self-control among weight control behavior in young people with overweight and obesity. Policy implications may include emphasizing that medical institutions and medical professionals must consider self-control as important in order to induce healthy weight control among individuals with overweight and obesity, a public health issue that is increasing worldwide. We believe that our study makes a significant contribution to the literature because the positive findings can be incorporated into nursing practice and as part of school or workplace health programs.

This manuscript has not been published or presented elsewhere in part or in entirety and is not under consideration by another journal. All study participants provided informed consent, and the study design was approved by the appropriate ethics review board. We have read and understood your journal’s policies, and we believe that neither the manuscript nor the study violates any of these. There are no conflicts of interest to declare.

Thank you for your consideration. I look forward to hearing from you.

Sincerely,

Hyun Kyeong Park

Department of Nursing, College of Nursing

Dongguk University Gyeongju Campus

123, Dongdae-ro, Gyeongju-si

Gyeongsangbuk-do, Korea 38066

Tel. 82-10-4286-3678

Fax. 82-54-770-2616

E-mail: sg3679@dongguk.ac.kr

Reviewer 3 Report

Comments and Suggestions for Authors

Many thanks to the authors for taking a very diligent approach to addressing each of the comments. There have been many significant improvements to the manuscript.

Consider whether the implications of the results could be explored better in the discussion. For instance, what other mediators should be explored (Line 286)? Always good to be cautious, but some claims are still rather general.

The aims are now much clearer but consider the balance provided to self-control and associated literature to the more “broad” literature.

Author Response

(The authors gave the same response as above.)
